# PI3Kδ Inhibition Potentiates Glucocorticoids in B-lymphoblastic Leukemia by Decreasing Receptor Phosphorylation and Enhancing Gene Regulation

**DOI:** 10.3390/cancers16010143

**Published:** 2023-12-27

**Authors:** Jessica A. O. Zimmerman, Mimi Fang, Miles A. Pufall

**Affiliations:** 1Division of Pediatric Hematology/Oncology, Stead Family Department of Pediatrics, Carver College of Medicine, University of Iowa, Iowa City, IA 52242, USA; jessica-zimmerman@uiowa.edu; 2Holden Comprehensive Cancer Center, University of Iowa, Iowa City, IA 52242, USA; mimi-fang@uiowa.edu; 3Department of Biochemistry and Molecular Biology, Carver College of Medicine, University of Iowa, Iowa City, IA 52242, USA

**Keywords:** glucocorticoids, chemotherapy, drug resistance, neoplasia-lymphoid leukemias, pediatric hematology/oncology

## Abstract

**Simple Summary:**

B-lymphoblastic leukemia (B-ALL) is the most common childhood cancer. Combination chemotherapies that include glucocorticoids (steroids) are effective in ~90% of children; however, the treatment is brutal. For children whose B-ALL relapses, cure rates remain <50% even with newer immunotherapies. Improvements in the current treatment are urgently needed. Because B-ALL response to steroids alone predicts if children will be cured, we sought to make steroids work better by adding another drug, idelalisib, to better treat B-ALL while reducing side effects. We show that this combination was effective in 90% of patients’ B-ALL cells. Idelalisib works in part by making the target of steroids, the glucocorticoid receptor, more active, particularly in B-ALL cells that responded poorly to steroids. The success of the combination of steroids and idelalisib in the lab indicates that adding idelalisib should be effective for most patients in the clinic, improving outcomes and opening the door to gentler treatments.

**Abstract:**

Glucocorticoids are the cornerstone of B-lymphoblastic leukemia (B-ALL) therapy. Because response to glucocorticoids alone predicts overall outcomes for B-ALL, enhancing glucocorticoid potency should improve treatment. We previously showed that inhibition of the lymphoid-restricted PI3Kδ with idelalisib enhances glucocorticoid activity in B-ALL cells. Here, we show that idelalisib enhances glucocorticoid potency in 90% of primary B-ALL specimens and is most pronounced at sub-saturating doses of glucocorticoids near the EC50. Potentiation is associated with enhanced regulation of all glucocorticoid-regulated genes, including genes that drive B-ALL cell death. Idelalisib reduces phosphorylation of the glucocorticoid receptor (GR) at PI3Kδ/MAPK1 (ERK2) targets S203 and S226. Ablation of these phospho-acceptor sites enhances sensitivity to glucocorticoids with ablation of S226 in particular reducing synergy. We also show that phosphorylation of S226 reduces the affinity of GR for DNA in vitro. We propose that PI3Kδ inhibition improves glucocorticoid efficacy in B-ALL in part by decreasing GR phosphorylation, increasing DNA binding affinity, and enhancing downstream gene regulation. This mechanism and the response of patient specimens suggest that idelalisib will benefit most patients with B-ALL, but particularly patients with less responsive, including high-risk, disease. This combination is also promising for the development of less toxic glucocorticoid-sparing therapies.

## 1. Introduction

Glucocorticoids, including dexamethasone and prednisone, are the cornerstone of chemotherapy regimens for B-lymphoblastic leukemia (B-ALL), the most common childhood cancer [1]. Although about 90% of patients with standard-risk B-ALL are cured with glucocorticoid-based therapies, only about 75% of patients with high-risk B-ALL are as fortunate. Because response to glucocorticoid therapy alone predicts overall outcomes for patients with B-ALL [2,3], enhancing GR activity is an attractive mechanism for improving outcomes, particularly in high-risk disease. The use of more potent or higher dose glucocorticoids is prevented by systemic toxicities, including osteonecrosis, myopathy, and pancreatitis, which significantly impact quality of life and may be life threatening [4,5,6,7,8]. Because glucocorticoids work cell-autonomously to kill B-ALL cells, one solution is to enhance GR activity specifically in B lymphocytes. We therefore sought to enhance both prednisone, which is frequently used in high-risk B-ALL induction regimens, and dexamethasone by targeting B-ALL-restricted proteins.

We, and others [9], identified PI3Kδ as one such promising target. PI3Kδ is a leukocyte-restricted kinase and a key component of the B-cell receptor (BCR) and interleukin-7 receptor (IL-7R) pathways in B cells. Using functional genomics, we found that knockdown or inhibition of PI3Kδ sensitizes B-ALL cell to glucocorticoids, likely by blocking signaling through the RAS/MAPK pathway [10]. Although PI3Kδ can activate the AKT pathway, which restrains glucocorticoid potency in some T-ALL [11], we found that AKT had no effect on glucocorticoid sensitivity in the B-ALL specimens tested [10]. These findings suggested that inhibition of PI3Kδ would be an effective way to enhance glucocorticoid potency specifically in B-ALL without increasing systemic potency and toxicity.

Idelalisib is a first-in-class, isoform-specific PI3Kδ inhibitor used in the treatment of relapsed chronic lymphocytic leukemia. Though effective, long-term use of PI3Kδ inhibitors has been associated with significant immune-mediated adverse effects, primarily through inhibition of regulatory T-cells [12]. Intermittent dosing has been shown to improve adverse effects while maintaining anti-tumor activity [12] and could therefore be used during glucocorticoid courses in B-ALL treatment regimens. Using a limited number of patient specimens and cell lines, we found that idelalisib synergistically enhances dexamethasone-induced B-ALL cell death [10], underscoring its potential. It is not known whether idelalisib also enhances prednisone potency, particularly in high-risk B-ALL.

Glucocorticoids work through the glucocorticoid receptor (GR), a ligand-activated transcription factor, to regulate thousands of genes, including those that induce cell death in B-ALL. Glucocorticoids regulate ~80 effector genes—genes whose regulation contributes to cell death. Effector genes include those involved in apoptosis (suppression of *BCL2* and upregulation of *BCL2L11* [13,14]) and B-cell development [10]. Testing a handful of these effectors, we found that idelalisib enhanced the regulation of some, but not all, of these effector genes, with different effects in different cell lines [10]. This suggested the model that idelalisib enhances glucocorticoid potency in different ways in different types of B-ALL. However, a more systematic analysis of the effect of idelalisib on glucocorticoid gene regulation in primary specimens is needed to test this model.

The potentiation of glucocorticoids by PI3Kδ inhibition could be a result of blocking GR phosphorylation. Consistent with our model, phosphorylation by the BCR/RAS/MAPK pathway modulates GR activity in a cell-type and gene-specific manner [15,16,17] and has been associated with glucocorticoid resistance [18]. ERK2 (MAPK1), the terminal kinase in the RAS/MAPK pathway, has been shown to phosphorylate GR at multiple sites, including S203, S211, and S226 [15,16], in other systems. Treatment of B-ALL cells with idelalisib is accompanied by a decrease in phosphorylation of GR at S203 [10], but may affect other sites as well. It is not clear which sites of MAPK phosphorylation affect GR activity in B-ALL and how they might do so in a cell- or gene-specific manner.

Here, we show that, as with dexamethasone [10], idelalisib enhances prednisolone-induced cell death in B-ALL cell lines and most primary patient specimens and is particularly synergistic at concentrations near the EC50 of dexamethasone and prednisolone. Counter to our previous model, we show that idelalisib enhances glucocorticoid-induced regulation of virtually all genes across almost all specimens tested. Consistent with this global effect on gene regulation, we show that phosphorylation of S226 decreases the affinity of GR for DNA, and preventing phosphorylation at S226 enhances glucocorticoid potency while significantly blunting synergy with idelalisib. We therefore propose that idelalisib potentiates glucocorticoids in part by reducing phosphorylation of GR at S226, enhancing DNA binding and regulation of effector genes.

## 2. Materials and Methods

### 2.1. Cell Viability Assays

B-ALL cell lines (NALM6 and RS4;11) and NALM6 phospho-GR mutants (GR-S203A and GR-S226A) were tested with combinations of prednisolone (Acros Organics, ThermoFisher, Waltham, MA, USA, #449470250) and the PI3Kδ inhibitor idelalisib (Gilead Sciences, Foster City, CA, USA). Viability was measured using PrestoBlue (ThermoFisher, Waltham, MA, USA, A13262). Synergy was evaluated using the Bliss synergy model in SynergyFinder 2.0 [19].

Primary specimens from children with newly diagnosed or relapsed B-ALL were obtained prior to initiating treatment after receiving informed consent (University of Iowa IRB protocol #201707711). Cells were obtained from either peripheral blood or bone marrow and isolated by Histopaque density gradient separation. Freshly isolated cells were used for all experiments involving primary specimens.

NALM6, SUP-B15, and RCH-ACV cells were tested with dexamethasone (Sigma-Alrich, St. Louis, MO, USA, D4902-1g) in combination with ERK1/2 inhibitor SCH772984 (SelleckChem, Houston, TX, USA, #S7101).

### 2.2. Gene Expression Analysis of NALM6 Cells and Primary Specimens

NALM6 cells were treated with vehicle, dexamethasone (5 nM or 50 nM), idelalisib (250 nM), or dexamethasone and idelalisib. Seven primary patient specimens (MAP010, MAP014, MAP015, MAP016, MAP019, MAP020, MAP031) were treated for 24 h with vehicle, dexamethasone (25 or 50 nM), prednisolone (25 or 50 nM), idelalisib (500 nM), combinations of dexamethasone and idelalisib, or combinations of prednisolone and idelalisib. Sequencing data were processed using R/Bioconductor and DESeq2 [20] with RUVSeq [21]. Code for the analysis is available in the Appendix A.

### 2.3. Protein Expression and Purification

The human GR AF1-DBD (27-506) polypeptide, containing most of the N-terminal AF1 region and the DNA binding domain (DBD), was expressed and purified as described in [22]. ERK2 was expressed and purified as described in [23].

### 2.4. Phosphorylation of GR-AF1-DBD and Purification

GR-AF1-DBD was phosphorylated with ERK2 (30 min at 30 °C). GR-AF1-DBD phosphorylated species were separated on a 5/5 MonoQ column (Cytiva, Marlborough, MA, USA) and run over a size exclusion column (Cytiva, Marlborough, MA, USA, Superdex 200).

### 2.5. Mass Spectrometry

GR-AF1-DBD +/− phosphorylation samples were reduced, alkylated, digested with trypsin, and purified with C18 stage tips (Pierce, Dallas, TX, USA, #87781) [24,25]. Peptides were separated by LC and analyzed on a QExactive HF Orbitrap (ThermoFisher, Waltham, MA, USA) mass spectrometer. Fully phosphorylated GR and singly phosphorylated GR were compared to unmodified GR-AF1-DBD using Scaffold 5.1.2 (Proteome Software Inc., Portland, OR, USA). Phospho-fragment frequencies were compared to the control to determine the level of phosphorylation at each site.

### 2.6. Electrophoretic Mobility Shift Assays (EMSA)

The dissociation constants for unmodified and phosphorylated GR-AF1-DBD fragments were measured by EMSA as described in [22] using a consensus GR site (5′-GTAC**GGAACA**TCG**TGTACT**GTAC-3′).

### 2.7. Phospho-GR Western Blotting

NALM6 cells were treated with vehicle, dexamethasone (5 nM or 1 μM), idelalisib 250 nM, or dexamethasone plus idelalisib for 24 h. Western blotting was performed as described in [10] using GR-S203P or GR-S226P rabbit polyclonal antibody (generously provided by the Garabedian Lab, New York University, New York, NY, USA) or GR IA-1 [10]. Changes in GR phosphorylation were determined as the ratio of phospho-GR to total GR compared to controls.

### 2.8. Phospho-GR Mutants by CRISPR

Cas9-RNPs were transfected into cells by electroporation (SF Cell Line 4D-Nucleofector™ X Kit S (Lonza, Basel, Switzerland, #V4XC-2032)). After 48–72 h, editing efficiency was checked by T7EI digest (IDT, Coralville, IA, USA, #1075931). Cells were single-cell sorted (Becton Dickinson Aria II) into 96 well plates. Positive clones were identified by extracting genomic DNA, PCR amplifying the region, Sanger sequencing of control, experimental, and reference PCR products, and analyzing by TIDER [26].

Additional details are available in the Appendix A.

## 3. Results

### 3.1. Inhibition of PI3Kδ Increases Prednisolone Sensitivity in B-ALL Cell Lines and Primary Patient Specimens

To determine whether idelalisib synergizes with prednisolone to induce B-ALL cell death similar to dexamethasone, we titrated both into two B-ALL cell lines (RS4;11, NALM6) and measured viability. RS4;11 cells were more sensitive to prednisolone (EC50 = 25 nM) than NALM6 cells (EC50 = 80 nM). To quantify the impact of the addition of idelalisib, we evaluated both overall and peak (i.e., highlighting the doses with strongest synergy) Bliss scores with SynergyFinder [19]. When combined with idelalisib, RS4;11 and NALM6 cells exhibited additivity based on overall Bliss score (RS4;11: 5.826 ± 0.88; NALM6: 3.055 ± 0.56), but both were highly synergistic at some prednisolone concentrations (peak Bliss score 23.30 for RS4;11; 23.79 for NALM6). The idelalisib concentration at peak synergy was similar in both cell lines (~1.5 μM), but the prednisolone concentration was around half of the EC50 (RS4;11 ~10 nM; NALM6 ~40 nM) (Figure 1A,B). This suggested that idelalisib is synergistic at prednisolone concentrations near the EC50 and lower than peak clinical concentrations (~4 µM for 60 mg/m^2^ prednisone dose).

We then evaluated prednisolone/idelalisib synergy by testing 20 primary B-ALL specimens (Appendix A). Most primary specimens were sensitive to prednisolone (EC50 = 6.5–71 nM). Three specimens were glucocorticoid resistant, including one with no response to glucocorticoids (MAP010: 105 nM; MAP021: 250 nM; MAP020 no response). We then compared the EC50 of each specimen to the viability of the specimen at the maximum concentration of prednisolone (10 μM) (Appendix A). Fourteen specimens had <30% viability, with the majority (12/14) from patients with negative end-of-induction minimal residual disease (MRD). Of the six specimens with >50% viability, four were MRD positive, one was MRD negative, and one was unknown. Thus, consistent with the literature [2,27], we find that prednisolone response ex vivo correlates well with patient response, particularly with respect to viability with high-dose prednisolone (Appendix A).

To account for both the prednisolone EC50 and viability of each specimen, we calculated the area under the curve (AUC) and compared these to overall and peak Bliss scores. Although no correlation reached statistical significance, a trend emerged that higher prednisolone AUC (i.e., higher glucocorticoid sensitivity) is associated with lower overall and peak Bliss scores (R = −0.34 and −0.37, respectively; Figure 1C). This suggests that glucocorticoid-resistant specimens derive more synergy with the addition of idelalisib to glucocorticoids.

We next attempted to identify a pattern predicting synergy between glucocorticoids and idelalisib, particularly through National Cancer Institute (NCI) risk grouping or cytogenetic features. An additive response of idelalisib with prednisolone was observed in 90% (18/20) of specimens based on overall Bliss score, including both NCI standard-risk and NCI high-risk specimens with favorable, neutral, and unfavorable cytogenetics. Overall antagonism was evident in the two remaining specimens with notable cytogenetic features—MAP010 (overall Bliss score −10.6 ± 1.1) with near haploid cytogenetics, and MAP018 (overall Bliss score −15.4 ± 2.4) with *BCR::ABL1*. Fourteen specimens exhibited a synergistic peak Bliss score (Figure 1D–G and Appendix A). As in the cell lines, synergistic peak Bliss scores were observed near or below the prednisolone EC50 in all specimens (including relapsed B-ALL specimen, MAP019) except for three, each of which had >50% viability at high prednisolone doses (MAP012, MAP020, MAP025). Unlike cell line models, peak synergy in primary patient specimens does not occur at a consistent idelalisib concentration. We were also able to treat seven specimens with dexamethasone in combination with idelalisib, producing similar results (Appendix A). Thus, synergy between idelalisib and glucocorticoids was evident across NCI risk and cytogenetic groups, including high-risk and relapsed specimens, though without a consistent pattern.

### 3.2. PI3Kδ Inhibition Induces Global Enhancement of Glucocorticoid-Induced Gene Regulation in NALM6 Cells

To understand the mechanism of idelalisib-enhanced cell death, we measured changes in gene regulation with dexamethasone and idelalisib in NALM6 cells by RNA-seq. We first identified differentially regulated genes (adjp ≤ 0.01) in response to low dexamethasone (5 nM, ~EC50), high dexamethasone (50 nM), idelalisib (250 nM), and the combination of low and high dexamethasone with idelalisib. Idelalisib induced up- and downregulation of 418 genes (Figure 2A). Low dexamethasone induced regulation of fewer genes (649) than high dexamethasone (3779). The combination caused a greater than additive effect in the number of genes regulated, particularly with low dexamethasone (2398 combination vs. 1067 separately) compared to high dexamethasone (4965 combination vs. 4197 separately). This indicates that the combination either induces regulation of new genes relative to either drug alone or enhances dexamethasone-induced gene regulation.

To distinguish between these models, we performed linear regression on genes regulated by dexamethasone plus idelalisib (Figure 2B,C). For genes regulated by low dexamethasone, idelalisib significantly enhanced both upregulation (*p* = 2 × 10^−10^) and downregulation (*p* = 1 × 10^15^) with an average enhancement of 17% (*p* < × 10^−12^). There was a more modest, but significant, effect of idelalisib with high dexamethasone (6%, *p* < 2.2 × 10^−16^). This supports the model that idelalisib better enhances gene regulation at glucocorticoid concentrations closer to the EC50 than at high concentrations.

To determine whether adding idelalisib causes regulation of new genes, we incorporated an interaction term in the differential gene expression model. Of 2398 genes regulated by low dexamethasone plus idelalisib, only 18 exhibited a greater than additive effect. Of 4965 genes regulated by high dexamethasone and idelalisib, 72 showed a greater than additive effect. This indicates that a minority of genes are synergistically or newly regulated by the combination, which we evaluate below.

### 3.3. Idelalisib Potentiates Glucocorticoid-Induced Cell Death by Enhancing Effector Gene Regulation

We first sought to determine whether idelalisib potentiates regulation of effector genes. We identified effector genes by integrating gene regulation data with the results of two large-scale gene knockdown screens in NALM-6 cells previously performed by our lab to identify genes impacting glucocorticoid-mediated cell death [10,28] (Figure 2D, Appendix A). Positive effectors were genes that contributed to dexamethasone-induced cell death (as measured in the screens) and were upregulated by dexamethasone, thereby enhancing cell death. Negative effectors were genes that impaired dexamethasone-induced cell death but were downregulated by dexamethasone, also enhancing cell death. Our highest confidence effector genes were significant in both versions of the screen [10,28] (*p* < 0.01).

As with most dexamethasone-regulated genes, idelalisib significantly enhanced up- and downregulation of effector genes with low dexamethasone (19%, *p* = 0.007) but only sporadically enhanced genes with high dexamethasone (Figure 2E). Only *LSS* and *MED13L* were newly or synergistically regulated upon addition of idelalisib. This indicates that the primary mechanism of glucocorticoid potentiation by idelalisib in NALM6 cells is global enhancement of gene regulation, not regulation of new genes relative to either drug alone.

### 3.4. PI3Kδ Inhibition Enhances Glucocorticoid-Induced Gene Regulation in Some Primary Patient Specimens

To test whether idelalisib enhances glucocorticoid potency by globally enhancing gene regulation in a more clinically relevant context, we performed RNA-seq in seven freshly isolated primary B-ALL specimens. Because two of the seven specimens (MAP010, MAP020) were resistant to glucocorticoids (>50% viability at 10 µM prednisolone) and exhibited regulation of very few genes (dexamethasone 41, prednisolone 8, adjp < 0.01, Appendix A), we restricted further analysis to glucocorticoid-responsive specimens.

In the five glucocorticoid-responsive specimens (MAP014, MAP015, MAP016, MAP019, MAP031), dexamethasone induced regulation of 1951 genes and prednisolone induced regulation of 146 genes (Figure 3A). Idelalisib alone only regulated 6 genes, but the addition of idelalisib increased the number of genes regulated by both dexamethasone (2027 genes) and prednisolone (390 genes). Interestingly, genes regulated by dexamethasone were less upregulated or downregulated by idelalisib, although the change was modest (2.4%, *p* = 2 × 10^−5^, Figure 3B and Appendix A). Genes regulated by prednisolone showed a trend toward enhanced regulation by idelalisib (4%) but this did not reach statistical significance (*p* = 0.24) (Figure 3C and Appendix A). Idelalisib also showed a similar pattern of enhanced expression as in NALM6 cells for effector genes (Appendix A). The concentrations of prednisolone used were near or below the EC50 for each of the specimens, but the dexamethasone concentrations were over 10 times higher than the EC50. This is consistent with the model that idelalisib better enhances gene regulation at glucocorticoid concentrations close to the EC50.

We then evaluated whether idelalisib enhanced gene regulation in specimens that responded synergistically (MAP015, MAP019) compared to those that responded additively (MAP014, MAP031). Counterintuitively, dexamethasone or prednisolone alone regulated more genes in additive (Figure 3D–F) compared to synergistic (Figure 3G–I) specimens (dexamethasone 6195 vs. 4015; prednisolone 1447 vs. 1115; adjp ≤ 0.01). Idelalisib alone regulated genes in additive specimens (298) but few in synergistic specimens (10). When comparing glucocorticoid-regulated genes shared by both groups, additive specimens demonstrated stronger gene regulation with dexamethasone, and to a lesser extent with prednisolone, compared to synergistic specimens (Appendix A). Additive specimens were also more sensitive to prednisolone alone based on viability compared to synergistic specimens (additive: MAP014 13%, MAP031 8%; synergistic: MAP015 25%, MAP019 27%). We therefore proposed that synergistic specimens are less glucocorticoid-sensitive and thus more amenable to glucocorticoid potentiation than additive specimens.

To test this, we examined how the addition of idelalisib changed the glucocorticoid-induced gene regulation in the additive and synergistic specimens. For additive specimens (Figure 3E and Appendix A), idelalisib modestly but significantly enhanced dexamethasone regulation of genes (1.3%, *p* = 0.001). The effect of idelalisib on prednisolone-induced gene regulation was more pronounced (Figure 3F and Appendix A) with an average enhancement of 11% (*p* < 2.2 × 10^−16^). Counter to the model, the addition of idelalisib to synergistic specimens led to genes being modestly but significantly less regulated with either dexamethasone (Figure 3H and Appendix A) (4%, *p* < 2.2 × 10^−16^) or prednisolone (Figure 3I and Appendix A) (5%, *p* = 1 ×10^−13^). The enhanced gene regulation in additive specimens may be due to a stronger effect on gene regulation by idelalisib alone (Figure 3D) compared to the synergistic specimens (Figure 3G). Examining the effector genes, although there is evidence that the regulation of some is enhanced by idelalisib in the synergistic specimens, the effect is not consistent as in NALM6 cells (Appendix A). We speculate that the genes with enhanced regulation by idelalisib contribute to synergy, but that the main synergistic effect is through other pathways. Increasing the number of samples analyzed may help resolve this ambiguity as the mechanism of idelalisib-induced glucocorticoid potentiation may differ in different B-ALL backgrounds.

### 3.5. GR Is Phosphorylated by ERK2 at Six Sites, Most Prominently S226

To understand the mechanism of idelalisib-enhanced gene regulation, we examined how idelalisib affects GR phosphorylation. We previously mapped MAPK1/ERK2, a key kinase in B-ALL transformation [29], downstream of PI3Kδ in the B-cell receptor pathway [10]. Inhibition of the RAS/MAPK pathway, and not the AKT/mTOR pathway, appeared to be the main driver of increased glucocorticoid sensitivity with PI3Kδ inhibition [10]. To test this, we measured synergy between an ERK1/2 inhibitor (SCH772984) and dexamethasone in three B-ALL cell lines (NALM6, SUP-B15, and RCH-ACV). The overall and peak Bliss scores were consistent with the combination of dexamethasone and idelalisib (Appendix A), supporting the model that ERK2 lies downstream of PI3Kδ.

We then examined how direct phosphorylation of GR by ERK2 [30] affects its activity. To identify relevant sites of ERK2 phosphorylation on GR, we expressed and purified the N-terminus and DNA binding domain (GR-AF1-DBD), which recapitulates the binding of full-length GR and retains the most commonly phosphorylated sites, and phosphorylated it with ERK2. This resulted in a hazy band shifted upward on an SDS-PAGE gel, indicating that GR-AF1-DBD was phosphorylated but as a mixture of species. We separated the differently phosphorylated species using strong anion exchange (MonoQ) into high- and low-mobility species (Figure 4A). Interestingly, the highly phosphorylated form of GR-AF1-DBD (GR-6P) eluted in the same volume as the unmodified GR-AF1-DBD over a size exclusion column, but the monophosphorylated form (GR-1P) eluted at a later volume. This indicates that GR-1P adopts a more compact conformation than the unmodified or GR-6P species (Appendix A).

Phosphopeptide mapping by mass spectrometry of GR-6P indicated that it is likely a mixture of species with six predominant sites of phosphorylation (Figure 4B). Two sites (S203 and S226) were previously reported as sites of ERK2 phosphorylation [31,32], but the others are rarely observed (S45, S267, and T288) or previously unreported (S148). The low-mobility species (GR-1P) was predominantly phosphorylated at a single site (S226).

We tested the effect of phosphorylation on GR activity by measuring the DNA-binding affinity by electrophoretic mobility shift assay (EMSA) for a consensus GR site (5′-GTAC**GGAACA**TCG**TGTACT**GTAC-3′). The affinity of GR-6P was modestly (50%) but significantly (*p* = 0.004) inhibited compared to the unmodified GR. Surprisingly, GR-1P was both substantially (~3x) and significantly (*p* < 0.0001) inhibited compared to unmodified and GR-6P (Figure 4C). This indicates that phosphorylation of GR by ERK2 not only inhibits GR binding to DNA, but the pattern of phosphorylation can have an important effect, with S226 phosphorylation as a key modification for directly regulating GR affinity.

### 3.6. Blocking Phosphorylation of GR S203 or S226 Increases Glucocorticoid Sensitivity and Contributes to Idelalisib-Induced Glucocorticoid Potentiation

To validate the importance of GR phosphorylation, we tested the effect of idelalisib on S226 phosphorylation in NALM6 cells (Figure 4D). Previous studies showed that idelalisib reduced S203 phosphorylation induced by high (1 µM) dexamethasone [10]. Here we observe that idelalisib alone significantly reduced S226 phosphorylation (*p* = 0.01). The addition of idelalisib to dexamethasone trended toward reducing phosphorylation at S226 but was more variable and not significant (Figure 4E). This indicates that idelalisib reduces phosphorylation of GR at both S203 and S226 but in different ways.

To test the importance of both S226 and S203, we generated phospho-acceptor mutants (GR-S203A and GR-S226A) in NALM6 cell lines using CRISPR. Blocking phosphorylation at S203 (Figure 4F) and S226 (Figure 4G) increased the sensitivity of NALM6 cells to prednisolone (EC50s: wild-type ~80 nM, GR-S203A ~70 nM, GR-S226A ~60 nM), indicating that phosphorylation at either site attenuates GR activity.

We then evaluated both phospho-acceptor mutants with a combination of prednisolone and idelalisib to determine the impact of blocked phosphorylation on synergy. Although the overall Bliss scores for prednisolone and idelalisib fall in the additive range (<10), the combination has synergy (>10) at certain concentrations. The GR-S203A cells had similar overall and peak Bliss scores compared to wild-type NALM6 cells (Figure 4H and Appendix A). However, the GR-S226A cells showed significantly decreased overall Bliss scores compared to wild-type and GR-S203A cells, resulting in an overall Bliss score not significantly different than zero (Figure 4H). Peak Bliss scores for GR-S226A cells were also lower than for wild-type NALM6 cells but with borderline significance (*p* = 0.06) (Figure 4H and Appendix A). This suggests that S203 and S226 both contribute to glucocorticoid sensitivity, but S226 plays a significant role in the synergy between idelalisib and glucocorticoids. Although overall synergy in GR-S226A cells is ablated, synergy persists in both GR-S203A and GR-S226A cells at some concentrations, indicating that other PI3Kδ targets likely impact idelalisib-induced glucocorticoid synergy (Figure 4I).

## 4. Discussion

Our previous work [10] identified PI3Kδ inhibition as a promising strategy for glucocorticoid potentiation in B-ALL for two main reasons: (1) small-molecule inhibitors against PI3Kδ are available for clinical use, and (2) *PIK3CD* expression is restricted to leukocytes, targeting glucocorticoid potentiation to these cells without increasing off-tumor toxicities. In this work, we show that the PI3Kδ inhibitor idelalisib potentiates both dexamethasone and prednisone. Prednisone is 6–10 times less potent than dexamethasone and is less effective in younger children with high-risk B-ALL, even adjusting for dose [33]. Patients over 10 years of age with high-risk B-ALL, who are more prone to relapse, receive prednisone during induction to reduce systemic toxicities [33]. Thus, the addition of idelalisib during induction has the potential to be well tolerated and improve outcomes for patients with high-risk disease.

The majority of these experiments were performed with primary patient specimens ex vivo. Because ex vivo testing has been shown to recapitulate both patient response and response in xenograft models [34,35], and we have observed synergy between idelalisib and dexamethasone in mouse immunocompromised xenograft models [10], we believe this approach to be a reasonable predictor of the response to this combination in patients.

The data presented here indicate that idelalisib may be most synergistic in B-ALL that is less sensitive to glucocorticoids, which predicts poorer outcomes for patients [28,36,37]. For high-risk B-ALL, glucocorticoids are administered in doses (10 mg/m^2^ for dexamethasone or 60 mg/m^2^ for prednisone) that result in a systemic concentration of ~700 nM for dexamethasone or ~4 µM for prednisone. We found that idelalisib is most synergistic as the glucocorticoid concentration approaches the EC50 and is more additive at high concentrations. As glucocorticoids are metabolized, their concentrations may approach the EC50 for B-ALL cells which are less sensitive to glucocorticoids. Idelalisib would have the most synergy with glucocorticoids in these B-ALL cells, increasing the chances of triggering cell death. It also suggests that the effective dose of glucocorticoids might be maintained over a longer time as the glucocorticoid is cleared, without increasing off-tumor toxicity.

Our previous studies suggested that idelalisib might work differently in different B-ALL backgrounds, perhaps limiting its use to only certain subtypes [10]. This was based on measuring synergy in a few cell lines and primary specimens and monitoring cell death effector genes using qPCR. When we measured the effect of idelalisib on the regulation of all genes in more specimens by RNA-seq, we found that idelalisib enhances the regulation of virtually all glucocorticoid-regulated genes rather than select glucocorticoid effector genes (Figure 2A,C,E). This enhancement of gene regulation is accompanied by reduced phosphorylation of GR at S203 and S226, the latter of which increases the affinity of GR for DNA. These data indicate that idelalisib is a general potentiator of glucocorticoid activity in B-ALL, and the mechanism of idelalisib is to increase the potency of GR in regulating all genes, including cell death effectors, in part by increasing DNA affinity (Figure 4I). This mechanism of generalized GR potentiation is consistent with the enhanced glucocorticoid potency observed in nearly all specimens with measurable glucocorticoid activity. We therefore propose that idelalisib can improve outcomes by potentiating glucocorticoids in high-risk B-ALL and may allow the reduction of glucocorticoid doses and their accompanying off-tumor toxicities without compromising outcomes.

We envision PI3Kδ inhibitors being combined with glucocorticoids for the treatment of B-ALL in two ways. First, a PI3Kδ inhibitor like idelalisib could be added to induction glucocorticoid therapy. During the first week of induction, idelalisib could be tested in combination with glucocorticoids using patient specimens ex vivo, an approach studied in multiple hematologic malignancies [34,38,39,40,41]. If idelalisib potentiates glucocorticoids ex vivo, idelalisib could be added for the remaining week(s) of induction glucocorticoid therapy. Second, idelalisib could be added to a lower dose of dexamethasone during the delayed intensification phase. This would enable the maintenance of on-target glucocorticoid potency while decreasing debilitating off-tumor effects of glucocorticoids, particularly the development of osteonecrosis. Both strategies could improve glucocorticoid potency during key phases of therapy while avoiding chronic administration of idelalisib, which has proven to be toxic [12].

Although B-ALL cells with GR phospho-acceptor mutations S203A or S226A are more sensitive to glucocorticoids, both retain overall synergy (S203A) or synergy with idelalisib at certain concentrations (S203A, S226A). This may be due to some or all of the other four sites of ERK2 phosphorylation, including one (S148) that has not been reported to be phosphorylated by any kinase. Testing each phosphorylation site individually and in combination is now feasible using CRISPR and could be performed to assess the importance of the patterns of GR phosphorylation on its activity. Unfortunately, although we were able to purify GR phosphorylated homogeneously at S226, isolation of other phosphorylated species is more challenging. This will likely prevent in vitro study of other GR phosphoforms.

PI3Kδ inhibition may exert its effects on glucocorticoid sensitivity by altering other downstream targets of ERK2 or by signaling through other pathways. For example, the PI3K/AKT pathway phosphorylates GR at S134, inhibiting its activity in T-ALL [11], and can phosphorylate and inhibit other apoptotic and antiproliferative pathways in B-cells [42,43,44]. We do not observe an effect on glucocorticoid sensitivity by knocking down AKT in NALM6 cells [10], but it may play a role in other B-ALL backgrounds. Further study of the downstream effects of idelalisib is needed to fully understand the complete mechanism of synergy with glucocorticoids.

Lastly, we speculate that idelalisib enhancement of glucocorticoid potency would occur in any B-cell malignancy derived from immature B cells that maintain BCR signaling, including DLBCL, CLL, and other lymphomas. This possibility remains to be tested.

## 5. Conclusions

We conclude that idelalisib will be an effective addition to existing combination therapies for children with B-ALL. Idelalisib enhanced glucocorticoid potency in 90% of patient specimens across risk stratifications and cytogenetic backgrounds. Idelalisib enhanced glucocorticoid potency in part by enhancing the regulation of all glucocorticoid-regulated genes, including effector genes that drive cell death. This enhancement can be linked to the suppression of GR phosphorylation at S226, relieving repression of DNA binding. The broad enhancement of glucocorticoid potency across patient specimens and the general enhancement of GR function provides strong evidence that idelalisib will be an effective way to enhance treatment efficacy and improve outcomes for most patients. Further, the addition of idelalisib is likely to be safe because it targets PI3Kδ, which is restricted to lymphoid cells, and will not increase systemic glucocorticoid toxicities.

## Figures and Tables

**Figure 1 cancers-16-00143-f001:**
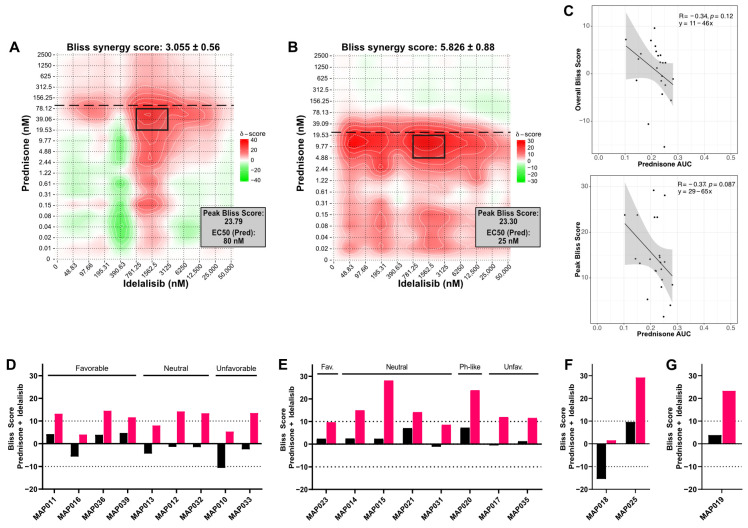
Prednisolone and idelalisib synergistically induce cell death in B-ALL cell lines and primary patient specimens in vitro. Evaluation of synergy in (**A**) NALM6 cells and (**B**) RS4;11 cells treated with the combination of prednisolone and idelalisib. A Bliss score greater than 10 indicates synergy, 10 to −10 indicates additivity, and less than −10 indicates antagonism. Overall Bliss score is at the top of each plot, and the peak Bliss score from the black outlined area is given in the box at the top right corner of each plot. Horizontal dashed line indicates the EC50 for prednisolone. (**C**) Correlations (Pearson coefficient R) of prednisolone AUC for patient specimens, NALM6, and RS4;11 cells with overall Bliss score (**top**) and peak Bliss score (**bottom**) demonstrate a trend toward decreased prednisolone sensitivity correlating with increased synergy with idelalisib, although neither has a *p* ≤ 0.05. (**D**–**G**) Overall (black bars) and peak (pink bars) Bliss scores for the combination treatment of prednisolone plus idelalisib in (**D**) NCI standard-risk specimens, (**E**) NCI high-risk specimens, (**F**) infant specimens, and (**G**) a relapsed specimen do not reveal an association between synergy and risk grouping or cytogenetic features. Cytogenetic features of these specimens are favorable/fav (*ETV6::RUNX1* or double trisomy), unfavorable/unfav (*KMT2A* rearrangement, iAMP21, or hypodiploidy), Ph-like (*P2RY8::CRLF2*), or neutral (all other cytogenetic features).

**Figure 2 cancers-16-00143-f002:**
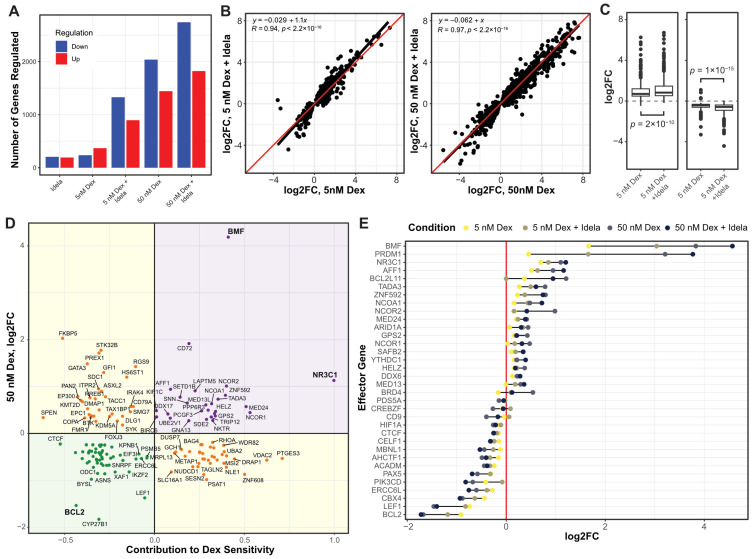
Idelalisib enhances dexamethasone regulation of effector genes in NALM6 cells. (**A**) The numbers of genes up- and downregulated (*p* ≤ 0.01) for NALM6 cells treated with combinations of dexamethasone (dex) and idelalisib (idela) for 24 h. (**B**) The log2 fold change in genes regulated by 5 nM dexamethasone is enhanced by idelalisib (**left**), whereas the enhancement by idelalisib at 50 nM dexamethasone (**right**) is less pronounced for most genes. The linear regression fit is a black line compared to the red line for no effect. (**C**) Box plots of upregulation (**left**) and downregulation (**right**) by 5 nM dexamethasone show significantly enhanced regulation by the addition of idelalisib. (**D**) Plot of the effect of each gene on dex-sensitivity (*x*-axis) versus regulation by 50 nM dexamethasone (*y*-axis). Positive effector (purple) and negative effector (green) genes are those whose regulation contributes to dex-induced NALM6 cell death, whereas buffering genes (yellow) oppose dex-induced cell death. (**E**) The log2 fold change of effector genes in response to combinations of dexamethasone and idelalisib. The vertical red line depicts no change in expression (log2FC = 0). In 2B and 2C, R = Pearson correlation coefficient. Idelalisib concentration is 250 nM in all experiments.

**Figure 3 cancers-16-00143-f003:**
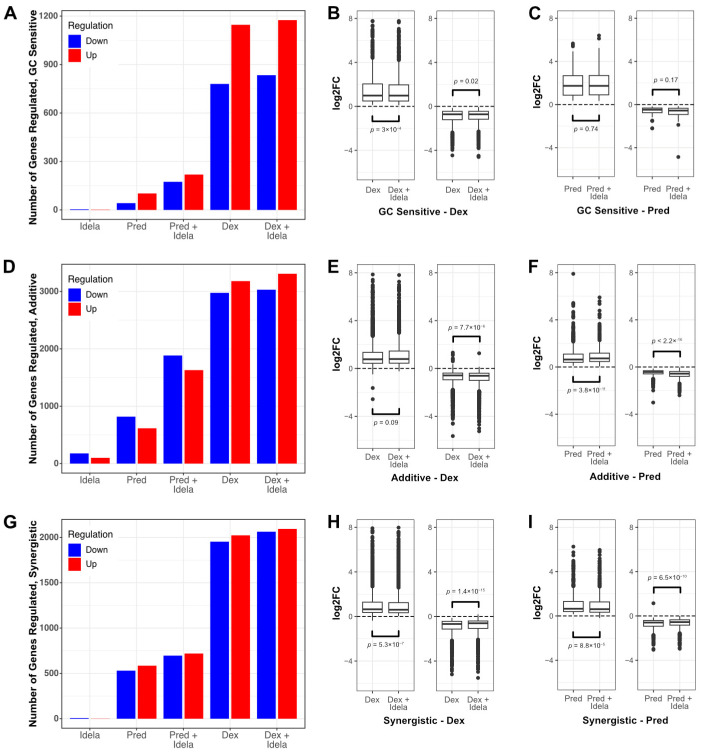
Idelalisib enhances glucocorticoid-induced gene regulation in primary patient specimens. (**A**) Overall number of genes upregulated (red) and downregulated (blue) in five glucocorticoid-sensitive primary patient specimens treated with idelalisib only (idela), prednisolone only (pred), prednisolone + idelalisib (pred + idela), dexamethasone only (dex), or dexamethasone + idelalisib (dex + idela). (**B**) Box plots of upregulation (left) and downregulation (right) in five glucocorticoid-sensitive specimens treated with dex vs. dex + idela. (**C**) Box plots of upregulated (**left**) and downregulated (**right**) genes in five glucocorticoid-sensitive specimens treated with pred vs. pred + idela. (**D**) Overall number of genes regulated in two primary patient specimens with an additive response (MAP014 Bliss score 3 and MAP031 Bliss score −5) to combination pred + idela treatment in viability assays at the concentrations used for RNA-seq. (**E**) Box plots of upregulated (**left**) and downregulated (**right**) genes in the two additive specimens treated with dex vs. dex + idela. (**F**) Box plots of upregulated (**left**) and downregulated (**right**) genes in the two additive specimens treated with pred vs. pred + idela. (**G**) Overall number of genes regulated in two primary patient specimens with a synergistic response (MAP015 Bliss score 22 and MAP019 Bliss score 14) to combination pred + idela treatment in viability assays at the concentrations used for RNA-seq. (**H**) Box plots of upregulated (**left**) and downregulated (**right**) genes in the two synergistic specimens treated with dex vs. dex + idela. (**I**) Box plots of upregulated (**left**) and downregulated (**right**) genes in the two synergistic specimens treated with pred vs. pred + idela. For all box plots, *p*-values for paired *t*-tests are reported.

**Figure 4 cancers-16-00143-f004:**
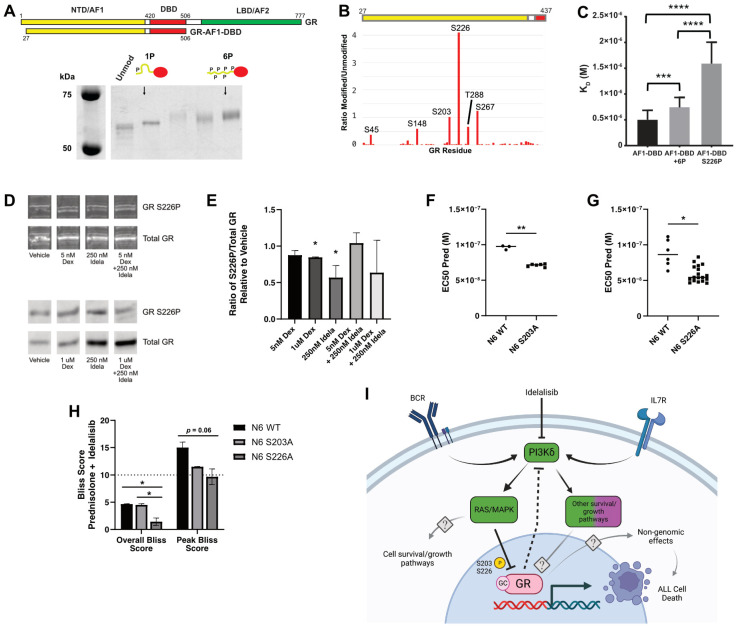
Phosphorylation of GR at S226 inhibits DNA binding, decreases phosphorylation of GR, and increases glucocorticoid sensitivity. (**A**) Purified GR-AF1-DBD was expressed and phosphorylated with ERK2 (NTD/AF1 = N-terminal domain/activation function 1, DBD = DNA binding domain, LBD = ligand binding domain/activation function 2). Phosphorylated species of GR-AF1-DBD were separated by strong anion exchange, isolating species with primarily one (1P) and six (6P) phosphates. (**B**) Mass spectrometry of phosphorylated GR-AF1-DBD-6P maps phosphorylation at six residues after ERK2 phosphorylation and isolation of GR-6P, including S203, with S226 being the most prevalent. (**C**) Unmodified GR-AF1-DBD binds with higher affinity than 6P. GR-1P (S226P) binding is more strongly inhibited. Dissociation constants (K_D_) were measured by electrophoretic mobility shift assays. Adjusted *p*-values from one-way ANOVA are 0.0036 (***) and <0.0001 (****). (**D**) Bands from representative Western blots illustrating phosphorylated S226 of GR in NALM6 cells treated with vehicle, dexamethasone (Dex) only, idelalisib (Idela) only, and dexamethasone with idelalisib (Dex + Idela) for 24 h. Both low-dose dexamethasone (5 nM, top) and high-dose dexamethasone (1 μM, bottom) were used. Phosphorylated S226 and total GR were blotted on the same membrane for each set of treatment conditions. (**E**) Quantification of the ratio of phosphorylated S226 to total GR normalized to vehicle control. Two biological replicates were performed for each condition except for idelalisib alone, which had 4 biological replicates. Phosphorylation of S226 is significantly reduced with both 1 μM Dex (*p* = 0.02) and 250 nM Idela (*p* = 0.01) compared to a normalized ratio of 1. (**F**) EC50 of prednisolone for CRISPR mutants with GR S203A compared to wild-type (WT) NALM6 cells. Two GR S203A clones were evaluated, with 3 biological replicates for each cell type. Welch’s *t* test *p* = 0.0010 (**). (**G**) EC50 of prednisolone for CRISPR mutants with GR S226A compared to wild-type (WT) NALM6 cells. Three GR S226A clones were evaluated, with 3 biological replicates per cell type. Welch’s *t* test *p* = 0.0111 (*). (**H**) Overall, Bliss scores for NALM6 S226A mutants are significantly decreased compared to WT NALM6 cells (*, *p =* 0.04) and NALM6 S203A mutants (*, *p =* 0.04). Peak Bliss scores for NALM6 S226A are decreased compared to NALM6 WT but not statistically significant (*p* = 0.06). NALM6 S203A mutants are not significantly different from NALM6 WT for either overall or peak Bliss scores. Bliss scores above 10 (dashed line) are considered synergistic. (**I**) Working model of idelalisib-induced glucocorticoid potentiation, where idelalisib inhibits PI3Kδ and subsequently reduces the inhibitory phosphorylation of GR at S226. BCR = B-cell receptor. IL7R = interleukin 7 receptor. GR = glucocorticoid receptor. GC = glucocorticoid. Created with BioRender.com.

## Data Availability

RNA sequencing data are available at Gene Expression Omnibus (NALM6 cells: GSE215385) and dbGaP (patient specimens: phs003085.v1.p1). Other original data are available upon request from the corresponding author.

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
