# Peer review of "PI3Kδ Inhibition Potentiates Glucocorticoids in B-lymphoblastic Leukemia by Decreasing Receptor Phosphorylation and Enhancing Gene Regulation"

_cancers, 2023, doi:10.3390/cancers16010143_

Round 1
Reviewer 1 Report
Comments and Suggestions for Authors
The authors have performed an expertly-done study with potentially-important implications for the treatment of B-ALL. The study investigated several different but relevant aspects of how PI3Kd inhibition might synergize with glucocorticoids: the dose-synergy relationship between glucocorticoids and idelalisib in cell lines and patient samples; effects of these drugs, alone and in combination, on glucocorticoid-induced changes in gene expression; and the changes in GR phosphorylation that appear to drive the synergistic effect of idelalisib on glucocorticoid sensitivity. I have only minor suggestions for how the manuscript might be improved.
1) More detail should be provided to support the interpretation of in vitro drug sensitivity testing of primary B-ALL samples. In particular:
What was the source of the samples: blood or bone marrow?
Were these all from previously-untreated patients?
Were these fresh or cryopreserved?
Was density gradient the only step in preparation of B-ALL cells? If so, what were the levels of purity, i.e., what was the degree to which cell types other than B-ALL were present?
What was the level of viability in the control (non-drug) cultures of B-ALL cells after 72 hours, when the viability results were measured? In other words, to what degree was there spontaneous apoptosis in the simple culture system?
2) All of the evaluations of drug sensitivity and gene expression were obtained from simple in vitro cultures, and are therefore cell-autonomous effects under artificial conditions. It remains to be determined whether these promising demonstrations of synergy, and the mechanism of changes in GR phosphorylation, will be observed in vivo, particularly with respect to what might be the immunologic effects of combining idelalisib with glucocorticoids in B-ALL. It would be a substantial undertaking for the authors to address this experimentally, even to the limited degree of using cell line xenografts in immunodeficient mice. It would be sufficient for the authors to discuss these caveats, and the need for further investigation.
3) Glucocorticoids are part of the standard treatment for other B-cell malignancies, notably diffuse large B-cell lymphoma (DLBCL) and multiple myeloma (MM), and particularly in the latter produce undesirable side effects. Idelalisib has been tested in DLBCL as a single agent for relapsed disease, with negative results. To my knowledge, the potential for idelalisib to enhance the efficacy and/or potency of glucocorticoids in DLBCL or MM has not been tested. The authors should mention this possibility.
Reviewer 2 Report
Comments and Suggestions for Authors
In the present manuscript (cancers-2752639), Jessica A.O. Zimmerman and colleagues investigate the mechanism through which PI3Kδ inhibition potentiates glucocorticoids in B-lymphoblastic Leukemia. Although the aim of this study is interesting and relevant as it would help to design less toxic glucocorticoid therapeutic regiments, the results do not clearly demonstrate how, mechanistically, PI3Kδ inhibition potentiates glucocorticoids in B-ALL. As such, I believe the manuscript would benefit from more experimental work before being suitable for publication in “Cancers”.
Main comments:
Page 8, line 299 – Are the effector genes enhanced by Idelalisib in dexamethasone and prednisolone treated B-ALL samples the same? Or the same effector genes enhanced in NALM6 cells?
Page 8, line 302 – It is not clear why the authors state that “idelalisib better enhances gene regulation at glucocorticoid concentrations close to the EC50” if they show that although “The concentrations of prednisolone used were near or below the EC50 for each of the specimens, but the dexamethasone concentrations were over 10 times higher than the EC50
Page 10, line 336 – The authors propose that the “synergistic specimens are less glucocorticoid sensitive and thus more amenable to glucocorticoid potentiation than additive specimens. However, “the addition of idelalisib to synergistic specimens led to genes being modestly but significantly less regulated with either dexamethasone or prednisolone” (line 344-346). How do the authors justify these findings? Since no regulation of effector genes was found (line 349), did the authors looked at other genes being regulated that could be responsible for idelalisib glucocorticoid potentiation in synergistic specimens?
Page 12, line 422 – When the authors tested the effect of the importance of idelalisib on GR phosphorylation in NALM6 cells they state that “idelalisib alone reduced S226 phosphorylation significantly (p = 0.01) but was more variable in combination with dexamethasone (Figure 4E).” and conclude that “This indicates that idelalisib reduces phosphorylation of GR at both S203 and S226 but in different ways.” Looking at Figure 4E I cannot understand how the authors can conclude that idelalisib in combination with dexamethasone reduces the phosphorylation of GR at S226 (with either dose of Dexa.). This is a key conclusion as it is used as part of the proposed mechanism of action of idelalisib glucocorticoid potentiation.
Page 13, line 434 – Why in this set of experiments the authors used the combination of prednisolone and idelalisib and in the previous ones the authors used the combination of dexamethasone and idelasib?
Minor comments:
Page 11 – Figure 4 legend not in bold.
Reviewer 3 Report
Comments and Suggestions for Authors
The manuscript by Zimmerman et al. titled "PI3Kδ Inhibition Potentiates Glucocorticoids in B-lymphoblastic Leukemia by Decreasing Receptor Phosphorylation and Enhancing Gene Regulation" presents compelling evidence on the potential benefits of idelalisib in enhancing glucocorticoid potency in B-ALL. However, there is a need for a more comprehensive rationale for PI3Kδ inhibition in B-ALL. The authors should include data on PI3Kδ and MAPK expression in either B-ALL cell lines or primary patient samples, comparing them to appropriate controls. This additional information will strengthen the foundation for the proposed mechanism.
While the combination of drug treatment appears promising, it is crucial to investigate whether this treatment regimen has any impact on the expression of IL7/IL7R in B-ALL cells. Considering the known role of IL7/IL7R in B-cell survival and proliferation, assessing its modulation could provide valuable insights into the broader effects of the drug combination on key signaling pathways in B-ALL.
The manuscript would benefit from an exploration of the in vivo effects of the drug combination. Understanding how this treatment strategy translates to an in vivo setting is essential for evaluating its clinical relevance. Including data on the effects of the drug combination in relevant animal models or patient-derived xenografts would strengthen the translational potential of the findings and enhance the overall impact of the study.
Addressing these concerns will not only enhance the rigor and completeness of the manuscript but also contribute to a more thorough understanding of the proposed therapeutic approach for B-ALL.
Round 2
Reviewer 2 Report
Comments and Suggestions for Authors
In the present manuscript (cancers-2752639), Jessica A.O. Zimmerman and colleagues investigate the mechanism through which PI3Kδ inhibition potentiates glucocorticoids in B-lymphoblastic Leukemia. The aim of this study is interesting and relevant as it would help designing less toxic glucocorticoid therapeutic regiments. Although I don’t think the results demonstrate clearly how, mechanistically, PI3Kδ inhibition potentiates glucocorticoids in B-ALL and strongly believe the manuscript would benefit from more experimental work, I do agree with the author’s claim in their report reply that “… the mechanism described, though not complete, provides a strong rationale for including idelalisib in treatment.” Having this in consideration as well as the changes done by the authors in the revised version of the manuscript, I think the manuscript can be now accepted in Cancers without the authors performing additional experiments as suggested.
Reviewer 3 Report
Comments and Suggestions for Authors
The authors effectively addressed all significant concerns, and I strongly endorse the publication of their work in the journal.